# Antimicrobial Activity of Individual Volatile Compounds from Various Essential Oils

**DOI:** 10.3390/molecules29081811

**Published:** 2024-04-16

**Authors:** Adriana Brandes, Mareshah Dunning, Jeffrey Langland

**Affiliations:** The Ric Scalzo Institute for Botanical Research, Sonoran University of Health Sciences, Tempe, AZ 85282, USA; a.brandes@sonoran.edu (A.B.); m.dunning@sonoran.edu (M.D.)

**Keywords:** essential oil, antibacterial, volatile compound

## Abstract

Interest in natural remedies has grown recently due to a variety of public health concerns such as microbial antibiotic resistance. This global health concern necessitates innovative approaches to combat bacterial infections. Building upon established therapeutic uses of essential oils, this research focused on the volatile constituents of essential oils. The volatile antimicrobial activity of these constituents was studied by employing a derivative of a modified disk diffusion assay for quantitative comparisons. This study emphasizes the significance and value of exploring natural compounds as alternatives to traditional antibiotics and provides insights into their mechanisms and applications in contending with bacterial pathogens.

## 1. Introduction

The discovery and widespread use of antibiotics, notably penicillin, revolutionized the treatment of bacterial infections [1]. Prior to penicillin’s discovery, treatments for infectious diseases relied on anecdotal remedies, including folk traditions involving aromatic herbs [2,3]. During the European Black Plague outbreak, caused by the bacterium *Yersinia pestis* in 1347, physicians resorted to unconventional techniques such as burning incense and aromatic herbs [3,4,5]. These historical practices underscore the contemporary interest in natural products as potential remedies for infectious diseases. The escalating issue of antibiotic resistance in recent years emphasizes the urgent need for alternative and complementary therapies [1,6,7]. The emergence of antibiotic-resistant bacterial strains has become a global health concern, necessitating the exploration of novel approaches to combat bacterial infections [8]. Within this context, this study investigated the antimicrobial activity of volatile constituents from essential oils.

While the use of essential oils for therapeutic purposes is well established, their antimicrobial potential through different modes of administration remains a topic of ongoing investigation. Past research has primarily focused on assessing the antimicrobial properties of essential oils in their liquid form when they directly interact with microorganisms [9]. This examination of antimicrobial effects, however, exerted by the volatile compounds released into the atmosphere as essential oils evaporate is relatively unexplored [10,11]. Notably, Maruzzella and Kienholz’s pioneering work in the mid-1900s involved a modified antimicrobial disk diffusion assay utilizing essential oil-saturated disks placed on inverted Petri dish lids [11,12,13]. As a modification of this design, the development of the reservoir diffusion assay in our previous research has allowed for the continued expansion of quantifying essential oils and their chemical constituent antimicrobial activity by allowing the zone of inhibition diameter to be measured [14]. Our previous research demonstrated that volatile constituents from cinnamon, rosemary, and thyme essential oils have potent antimicrobial efficacy [14]. Based on these previous findings and the established value of whole essential oils, we are continuing our research into individual compounds present in essential oils.

Essential oils contain bioactive compounds, such as terpenes, terpenoids, and phenylpropanoids, all of which have gained significant attention in recent years for their antimicrobial properties and are increasingly explored as potential solutions to combat antibiotic resistance [15,16,17,18,19]. Previous reports have demonstrated essential oil antimicrobial activity against various microorganisms, including *Staphylococcus aureus*, methicillin-resistant *Staphylococcus aureus* (MRSA), *Bacillus cereus*, *Escherichia coli*, and various fungal species [16,17,18]. Commonly, Gram-positive bacteria are more susceptible to terpenes than Gram-negative ones; it has been proposed that the lipophilic structure of essential oil constituents disrupt cell membranes leading to antimicrobial effects, although specific mechanisms remain unknown [17,20,21]. Recently, a synergy study was conducted exploring compounds in *Mentha piperita* essential oil, showing that these terpenoids interact synergistically with common antibiotics, augmenting their effectiveness [22]. Despite the strong evidence of antimicrobial activity, there remains a gap in research regarding the evaporative properties of essential oils against these pathogens. To address this gap, this study aimed to utilize a reservoir diffusion assay to further elucidate the volatile antimicrobial activity of the bioactive compounds present in essential oils.

## 2. Results

Based on our previous studies, *Streptococcus pyogenes* served as the model organism to assess the antimicrobial activity of specific aromatic constituents from rosemary, cinnamon, thyme, tea tree, and wintergreen essential oils in incremental amounts. In previous research and repeated here, rosemary, cinnamon, thyme, and tea tree were effective against *S. pyogenes,* while lemon and wintergreen were ineffective [14] (Figure 1). Rosemary and thyme EO showed a dose-dependent increase in the zone of inhibition with thyme reaching a maximal zone of inhibition from 20 to 80 µL/mL. For thyme, a larger zone of inhibition was observed for the 40 µL/mL dose compared to the 80 µL/mL dose, but this difference was not statistically significant and was likely due to experimental variation (Figure 1 and Figure 2). For cinnamon EO, only a minor dose dependence in the zone of inhibition was observed. This may be due to a limitation on how far the active constituents can diffuse into the air from the cylinder. This research further validates the activity of the aromatic constituents present in these essential oils against the model organism, *S. pyogenes*.

Since essential oils contain a variety of volatile terpenes and terpenoids, we next assessed the antimicrobial activity of the individual compounds. Based on gas-chromatography/mass-spectroscopy (GC-MS) analysis of the active antimicrobial essential oils, the amount of specific compounds within each essential oil were identified. Compounds with >5% concentration were selected as the major compounds to be tested for antimicrobial activity (Table 1). In addition, recreated essential oil blends were formulated that contained the compounds with >5% concentration combined in the same ratios as the original essential oil.

Tea tree oil showed a moderate range of antimicrobial activity against *S. pyogenes*. The major compounds present in tea tree essential oil, including the monoterpenes α-Terpinene, Terpinen-4-ol, and γ-Terpinene, showed low to moderate activity, each closely matching the zone of inhibition of the whole oil (Figure 2A). The recreated blend displayed increased antimicrobial efficacy compared to the whole oil or each of the individual monoterpenes at the highest concentration tested (Figure 2A). These results may suggest potential synergism between the individual compounds in the recreated blend that was not observed in the whole essential oil.

Cinnamon essential oil demonstrated high levels of volatile antimicrobial activity against *S. pyogenes* (Figure 2B). The primary compound of cinnamon oil, trans-Cinnamaldehyde, represents 45% of the composition of the oil. When tested alone, the trans-Cinnamaldehyde demonstrated similar to slightly higher volatile antimicrobial activity to that of the whole essential oil (Figure 2B). β-Caryophyllene and trans-Cinnamyl acetate, the other two most abundant compounds in cinnamon oil, contributed no antimicrobial activity. The recreated blend of cinnamon oil’s main constituents, trans-Cinnamaldehyde, β-Caryophyllene, and trans-Cinnamyl acetate, showed reduced activity compared to the whole oil or trans-Cinnamaldehyde alone (Figure 2B). These results may suggest that trans-Cinnamaldehyde was the major antimicrobial constituent and that the other constituents, β-Caryophyllene and trans-Cinnamyl acetate, may inhibit or reduce the activity of the trans-Cinnamaldehyde.

Rosemary oil exhibited dose-dependent volatile antimicrobial activity with high antimicrobial activity at its highest concentration (Figure 2C). The major compounds for rosemary oil include (+) α-pinene, (−) α-pinene, (+)-camphor, (−)-camphor, and 1,8-cineole. Each of these individual compounds exhibited no antimicrobial activity or weak antimicrobial activity (Figure 2C). In addition, the recreated blend of the major compounds also did not display any detectable volatile antimicrobial activity (Figure 2C). These results may suggest that the volatile antimicrobial activity of the whole rosemary essential oil was due to a minor constituent (<5%) present in the oil.

Whole thyme essential oil exhibited the most potent volatile antimicrobial activity (Figure 2D). The whole oil contains three major compounds: para-Cymene, γ-Terpinene, and Thymol. Thymol, a phenolic constituent, is the most abundant compound, comprising 40% of the thyme essential oil. Thymol demonstrated the highest inhibition among the three major compounds with antimicrobial activity higher than that of the whole essential oil (Figure 2D). Para-Cymene and γ-Terpene represented low to negligible activity. The recreated blend had activity similar to that of the whole essential oil (Figure 2D). These results support that Thymol was likely the major active constituent in thyme essential oil and that the other constituents may slightly inhibit the activity of Thymol.

## 3. Materials and Methods

Essential Oils. The essential oils examined were *Melaleuca alternifolia* (Tea Tree), *Rosmarinus officinalis* (Rosemary), *Cinnamomum zeylanicum* (Cinnamon), *Thymus vulgaris* (Thyme), and *Gaultheria fragrantissima* (Wintergreen). For consistency, the essential oils were sourced from one vendor, dōTERRA (Pleasant Grove, UT, USA). GC-mass-spectrometry conducted by a third party identified individual compounds within each essential oil. Gas chromatography analysis was carried out using a ZB5 column (60 m length × 0.25 mm inner diameter × 0.25 μm film thickness) with a Shimadzu GCMS-QP2010 Ultra instrument. Experimental conditions included a carrier gas of Helium at 80 psi, a temperature ramp of 2 °C per minute up to 260 °C, a split ratio of 30:1, and a sample preparation involving a 5% *w*/*v* solution with Dichloromethane. Gas chromatography profiles, along with their respective lot numbers, are accessible at http://sourcetoyou.com (accessed 1 January 2024), and further details are available in Table 1.

Chemical Compounds. Gas-Chromatography/Mass-Spectroscopy (GCMS) reports indicate the specified amounts of compounds within each essential oil. Compounds with >5% concentration were selected as the major compounds to be tested (Table 1). The major chemical compounds within each essential oil required for this study were obtained from Sigma-Aldrich [(+)-Camphor, (−)-Camphor, Cinnamyl acetate, trans-Cinnamaldehyde, p-Cymene, 1,8-Cineole, Linalool, (+)-α-Pinene, α-Terpinene, γ-Terpinene, and Terpinen-4-ol] and Chromadex [Thymol and β-Caryophyllene]. The major chemical compounds in each essential oil were tested individually and combined in similar ratios to make a recreated essential oil blend.

Antimicrobial sensitivity assay. The reservoir disk diffusion assay, designed in our previous research study was replicated to evaluate the volatile antimicrobial properties of the essential oils and constituents in a closed environment. Custom glass cylinders were designed to fit into the center of brain heart infusion (BHI) Petri dishes. These cylinders were 10 mm in diameter and height. Forty-eight-hour bacterial broth cultures using *Streptococcus pyogenes* (ATCC 12344) were used to inoculate the surface of BHI Petri dishes. A center plug of agar was removed, and a sterile glass cylinder was filled with either undiluted essential oil or individual compounds at the following amounts: 0 μL, 10 μL, 20 μL, 40 μL, and 80 μL. The individual compounds were prepared in a 1:1 ratio with 95% ethanol; 95% ethanol was used as a negative control and demonstrated no volatile antimicrobial activity. For the terpenes tested from each essential oil, the whole essential oil was used as a positive control and for comparison of possible increased or decreased activity of the individual compounds. Petri dishes were incubated in a sealed container for 48 h at 37 °C. After 48 h of incubation, the zone of inhibition (diameter) was measured. All experiments were done in triplicate.

Statistical analysis. Statistical analysis was performed using paired *t*-test. Statistically significant deviation of the various compounds and essential oils relative to the control treated (EtOH alone) with the *p*-value corresponding to the number of asterisks: * *p* = 0.01–0.05, ** *p* = 0.001–0.01, and *** *p* < 0.001.

## 4. Discussion

Essential oils consist of a complex blend of terpenes, terpenoids, phenolics, and other organic constituents responsible for their distinctive aromas, flavors, and biological activities [23,24]. Terpenes, in particular, play a significant role in the effectiveness of these oils [25]. Terpenes, which form the backbone of many essential oil constituents, are built from isoprene units, fundamental five-carbon molecules (C5H8) that serve as the building blocks for these larger structures. Isoprene units combine to form various types of terpenes: monoterpenes (10 carbons), diterpenes (20 carbons), sesquiterpenes (15 carbons), and triterpenes (30 carbons). This hydrocarbon structure gives terpenes their lipophilic nature and the addition of functional groups (alcohols, ketones, aldehydes, and esters, etc.) can affect their properties, such as solubility, antimicrobial activity, and other biological effects [26].

Previous studies have established that crude essential oils demonstrate stronger antimicrobial activity compared to their isolated constituents [27,28]. This is especially evident with rosemary oil, which contains 1,8-Cineole, α-Pinene, and Camphor as its primary components. Despite research illustrating that these individual compounds possess antimicrobial activity through direct application, 1,8-Cineole and Camphor had no volatile antimicrobial activity and α-Pinene had low volatile activity in this study [29,30]. In addition, a reconstructed blend of these components in the same ratios as the whole essential oil had no detectable volatile antimicrobial activity. The whole essential oil displays high activity, suggesting that other more minor constituents present in the oil are likely involved in the activity observed. Notably, research has shown that adding hydroxyl and other oxygenated groups to 1,8-Cineole through oxidation reactions improves its antimicrobial properties [31]. This modification serves to improve its solubility, membrane permeability, and interaction with microbial cells, thereby extending its bactericidal effects across a wider array of microorganisms [31,32].

Contrary to the belief that the whole essential oil is superior to individual constituents, some isolated essential oil compounds demonstrated high antimicrobial activity. Thymol has been identified as a particularly potent antibacterial agent, a finding that is well supported by the literature [27,33,34]. This discovery is in line with our results, wherein Thymol displayed the greatest activity. Interestingly, Thymol and Carvacrol, the predominant constituents in thyme and oregano essential oils, respectively, have been shown to demonstrate significant antibacterial efficacy against various antibiotic-resistant bacteria [20]. These closely related phenolic compounds are isomers, having the same molecular formula but differing in atom arrangements. The most frequently reported mechanism of antibacterial action for both Thymol and Carvacrol involves the disruption of bacterial membranes, leading to lysis and leakage of intracellular contents, ultimately resulting in bacterial death [35,36]. Studies have shown that both Thymol and Carvacrol’s antimicrobial activity is higher than that of other volatile compounds present in essential oils due to their hydrophobicity and the presence of a free hydroxyl group and phenol moiety [35,37,38]. In comparison, para-Cymene, which shares a similar chemical structure with Thymol and Carvacrol but lacks a free hydroxyl group, demonstrated no antimicrobial activity in our assays, underscoring the significance of this functional group. The presence of oxygenated moieties, as demonstrated by oxidized 1,8-Cineole derivatives, significantly enhances the antimicrobial activity of compounds, likely due to their proposed mechanism of membrane destabilization leading to cell death [31,32,35].

Terpinen-4-ol, a key component of tea tree oil, exhibits moderate antimicrobial activity. Terpinen-4-ol is slightly different from the structure of Thymol and Carvacrol. While it retains the ten-carbon skeleton typical of monoterpenes, it does not have a phenolic ring. Instead, Terpinen-4-ol has a hydroxyl group at the fourth carbon of a cyclic monoterpene structure [39]. The presence of this hydroxyl group attached to an aromatic ring allows for electron delocalization over the system, enhancing its ability to interact with biological targets [35]. The moderate biological activity observed in this study was likely due to the presence of the hydroxyl group which can participate in hydrogen bonding and other chemical interactions [40]. Tea tree oil also contains γ- and α-Terpinene, both cyclic monoterpenes, which exhibited low to moderate antimicrobial activity, respectively. These three compounds together likely contribute to the volatile antimicrobial activity observed with the whole tea tree essential oil. Interestingly, when these three compounds were blended together in the same ratios as the whole oil, increased antimicrobial activity was observed. This may suggest that other constituents present in the whole tea tree oil may inhibit or reduce the activity of these three compounds.

Cinnamon oil is a rich source of phenylpropanoids [41]. Phenylpropanoids are derived from amino acid precursors, leading to a different set of structural and functional characteristics compared to terpenes [42,43]. Trans-Cinnamaldehyde demonstrated high volatile antimicrobial activity with its effectiveness likely due to the delocalization of electrons across the double bonds of the benzene ring and the aldehyde functional group, making it an effective electrophile [44,45]. In contrast, trans-Cinnamyl acetate, another major compound, did not demonstrate antimicrobial activity, despite its structural relation to trans-Cinnamaldehyde. The key difference lies in the substitution of the aldehyde group with an ester functional group in Cinnamyl acetate. This modification significantly alters the molecule’s electronic properties, particularly its capacity for electron delocalization [43,46]. Previous research has shown that essential oil constituents can act synergistically, interacting with different targets of the cell phospholipid membrane to achieve activity due to their slight differences in structure [19,47]. With cinnamon oil, when a blend of the three major compounds, trans-Cinnamaldehyde, trans-Cinnamyl acetate, and β-Caryophyllene was tested, a substantial decrease in antimicrobial activity was observed suggesting negative inhibition between these molecules.

## 5. Conclusions

This study explored the volatile antimicrobial activity of four essential oils (tea tree, cinnamon, rosemary, and thyme) and their predominant constituents against the Gram-positive bacterium *S. pyogenes*. It revealed significant aerosolized antimicrobial activity in key bioactive constituents, particularly those with free-hydroxyl groups like thymol. Interestingly for rosemary, while individual compounds were effective, the whole essential oil demonstrated greater efficiency in killing bacteria, likely suggesting synergistic properties of minor constituents within the complex oil matrix. These findings underscore the importance of analyzing individual constituents versus whole oils for developing potent and cooperative antimicrobial formulations and highlight the potential of aerosolized essential oils for disinfection and air purification. This research emphasizes the need for further exploration into the antimicrobial activity of essential oils and their constituents. Future studies are needed to investigate the broad-spectrum antimicrobial activity of these agents against both Gram-positive and Gram-negative pathogens, to further elucidate the underlying mechanisms of action against bacteria, and to develop potential applications to combat infections and potential antibiotic resistance.

## Figures and Tables

**Figure 1 molecules-29-01811-f001:**
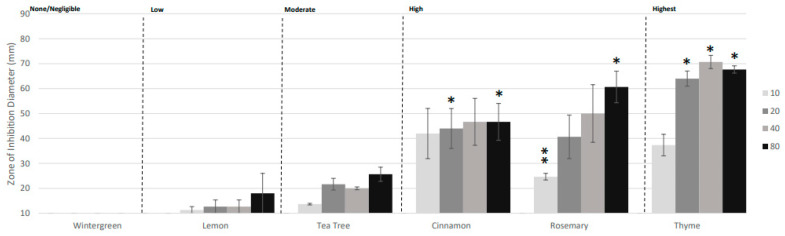
Volatile antibacterial activity of essential oils against *Streptococcus pyogenes*. Bacterial zone of inhibition assays were performed measuring the activity of volatile constituents from *Gaultheria fragrantissima* (Wintergreen), *Citrus limon* (Lemon), *Melaleuca alternifolia* (Tea Tree), *Cinnamomum zeylanicum* (Cinnamon), *Rosmarinus officinalis* (Rosemary), and *Thymus vulgaris* (Thyme) essential oils. The results are displayed with a color gradient from light to dark grey to represent the increasing essential oil concentrations, in the order of 0 μL, 10 μL, 20 μL, 40 μL, and 80 μL, respectively. The antimicrobial efficacy was assessed based on the zone of inhibition diameter and categorized as none (<10 mm), negligible (10–15 mm), low (15–30 mm), moderate (30–50 mm), high (50–70 mm), and highest (>70 mm). Error bars denote the standard deviation derived from three separate experiments. Statistically significant deviation of the various essential oils relative to an EtOH control are indicated with asterisks: * *p* = 0.01–0.05; ** *p* = 0.001–0.01.

**Figure 2 molecules-29-01811-f002:**
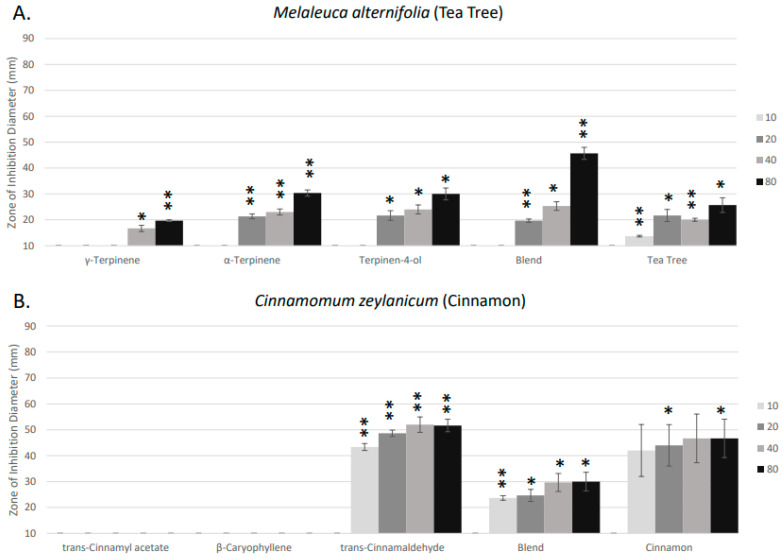
Antibacterial activity of essential oil volatile constituents against *Streptococcus pyogenes*. Bacterial zone of inhibition assays were performed measuring the activity of various volatile compounds and recreated compound blends derived from (**A**) *Melaleuca alternifolia* (Tea Tree), (**B**) *Cinnamomum zeylanicum* (Cinnamon), (**C**) *Rosmarinus officinalis* (Rosemary), and (**D**) *Thymus vulgaris* (Thyme). The results are displayed with a color gradient from light to dark grey to represent the increasing essential oil concentrations, in the order of 0 μL, 10 μL, 20 μL, 40 μL, and 80 μL, respectively. The antimicrobial efficacy was assessed based on the zone of inhibition diameter and categorized as none (<10 mm), negligible (10–15 mm), low (15–30 mm), moderate (30–50 mm), high (50–70 mm), and highest (>70 mm). Error bars denote the standard deviation derived from three separate experiments. Statistically significant deviation of the various isolated compounds and essential oils relative to an EtOH control are indicated with asterisks: * *p* = 0.01–0.05; ** *p* = 0.001–0.01; *** *p* < 0.001.

**Table 1 molecules-29-01811-t001:** Constituents present in essential oils. Gas Chromatography/Mass Spectroscopy were completed for each essential oil by Essential Oil University. This table lists the chemical constituents present in each essential oil within the following ranges: 5–10%, 10–20%, 20–30, and >30%.

	Lot #	Plant Part	5–10%	10–20%	20–30%	>30%
Wintergreen	211901A					Methyl Salicylate
Tea Tree	2222711A	Leaf	ND	α-Terpinene	γ-Terpinene	Terpinen-4-ol
Cinnamon	171326A	Bark	β-Caryophyllene	trans-Cinnamyl acetate	ND	trans-Cinnamaldehyde
Rosemary	171319A	Flower/Leaf	ND	α-Pinene Camphor	ND	1,8-Cineole
Thyme	222369A	Leaf	ND	γ-Terpinene para-Cymene	ND	Thymol

## Data Availability

All data are contained within the article.

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
