# Peer review of "Antimicrobial Activity of Individual Volatile Compounds from Various Essential Oils"

_molecules, 2024, doi:10.3390/molecules29081811_

Round 1

Reviewer 1 Report

Comments and Suggestions for Authors

In this article, Brander et al. reviewed the antimicrobial activity of individual volatile compounds from various essential oils. The overall study focuses on the importance and value of exploring natural compounds as alternatives to traditional antibiotics, providing insights into their mechanisms and applications in combating bacterial pathogens. Although the application suggests that this is a review article, the structure of the manuscript appears to resemble that of a research article. I recommend exercising control over this aspect.

The researchers selected the following essential oils obtained from the same vendor: Melaleuca alternifolia (Tea Tree), Rosmarinus officinalis (Rosemary), Cinnamomum zeylanicum (Cinnamon), Thymus vulgaris (Thyme), and Gaultheria fragrantissima (Wintergreen). Considering the wide profile of the materials, this selection is important for the field since it represents the major sources. The reservoir disk diffusion assay was employed to determine the antimicrobial sensitivity. The overall study is well planned, and the results are significant for the field due to the growing interest in pharmaceuticals. Despite these strengths, several points need to be addressed before acceptance:

-Please verify the type of the paper.

-The Introduction section should be extended to discuss the importance of these compounds in light of recent literature.

-Statistical analysis is missing throughout the paper. Appropriate analysis should be added.

-Figure 1 could be updated to include letters indicating the statistical differences between the groups, making it easier for readers to interpret.

-Similar updates should be made to Figure 2.

-These data are crucial for future studies. Please enhance the conclusion section by emphasizing future directions.

-The Conclusion section lacks a proper comparison of existing literature with the findings

Reviewer 2 Report

Comments and Suggestions for Authors

Comments to authors

In the manuscript titled " Antimicrobial Activity of Individual Volatile Compounds from Various Essential Oils", Brandes et al., studied ten volatile components from essential oils. Antimicrobial activity is measured using the reservoir disk diffusion assay. This study emphasizes exploring natural oils as alternatives to traditional antibiotics. Finally, they propose interaction mechanisms for the growth inhibition of bacterial pathogens. The introduction does not present the antecedents that support their investigations, so I consider that the introduction section must be rewritten and/or reformulated to support their investigation in the best way possible. For example, present studies of essential oils versus antibiotics or essential oils mixed with antibiotics.

Questions for the authors

·       The reservoir disk diffusion assay does not consider possible synergies or antagonisms between the components of the essential oil, why consider ethanol and rosemary oil as controls?

·       The viscosity of the oils is variable, and this will affect the diffusion in the agar. Therefore, the inhibition halo generated by each essential oil could be biased by diffusion problems. What considerations did you take to avoid misinterpretation?

·       Why did they only use gram-positive bacteria and not gram-negative bacteria?

·       In general, terpenes have a low polarity depending on the functional groups, and this type of molecule hardly interacts with gram-positive bacteria.

·       In this sense, what is the diffusion mechanism through the cell wall?

·       Can terpenes pass through the cell wall of Gram-positive bacteria? 

Additionally, below, please find several comments to the authors on specific parts of the manuscript.

Abstract, lines 7-8: “The general interest in natural remedies has increased due to the public’s distrust in pharmaceuticals and the presence of escalating health challenges such as microbial antibiotic resistance." Rewrite this paragraph.

Results and discussion, page 5 “Studies have shown that the antimicrobial activity of both thymol and carvacrol is greater than that of other volatile compounds present in essential oils due to the presence of the free hydroxyl group, hydrophobicity, and the phenol moiety.” Rewrite this paragraph, you are just using one reference.

Results, lines 169–173 “In comparison, para-Cymene, which shares a similar chemical structure with Thymol and Carvacrol but lacks a free hydroxyl group, demonstrated no volatile antimicrobial activity in our assays, underscoring the significance of this functional group. The presence of oxygenated moieties, as demonstrated by oxidized 1,8-Cineole derivatives, significantly enhances the antimicrobial activity of compounds. According to this paragraph, the hydroxyl groups present in terpenes are responsible for the antimicrobial activity. How do these interact with the cell membrane of bacteria?

Figure 1. When they used 40 and 80 μl of cinnamon essential oil in the antimicrobial evaluation, the halos were very similar. What is this behavior due to?

Figure 1. In the evaluation of thyme essential oil, why was a higher inhibition zone generated using 40 μl of essential oil compared to 80 μl?

Figure 2. In the evaluation of thyme essential oil, why was a higher inhibition zone generated using 40 μl of essential oil compared to 80 μl?

The conclusion is very extensive, I advise the authors that the current conclusion should be in the results and discussion section.

In general, the antimicrobial activity of the components present in essential oils is associated with whether they have hydroxyl groups. Steric hindrance could affect the type of essential oil-bacteria interaction, viscosity, and other functional groups.

Comments on the Quality of English Language

Dear authors, a moderate revision of the English language is required in your manuscript.

Reviewer 3 Report

Comments and Suggestions for Authors

This is a well written review based on a sound design. The elucidation of differences in antimicrobial activity between different essential oils, purified major components and a blend of the major compounds has been clearly demonstrated. The work using Strep pyogenes opens up avenues for future work with additional medically important bacteria.

My only suggestion for improvement is to incorporate most of what is presented under ‘Conclusions’ into a Discussion section. Conclusion would then be just a succinct essence of the review, in one or two paragraphs.

Round 2

Reviewer 1 Report

Comments and Suggestions for Authors

The authors significantly improved the manuscript and responded all the comments.

Author Response

Response not applicable.  Reviewer stated "significantly improved the manuscript and responded to all comments."

Reviewer 2 Report

Comments and Suggestions for Authors

It is important to clarify in the evaluation of thyme essential oil, why was a larger zone of inhibition generated using 40 μl of essential oil compared to 80 μl? (Figure 1 and 2).

As advice for your future investigations, consider possible diffusion problems in the evaluation of the antimicrobial properties of oils and their components. Would the behavior of the oils and their components be like gram-negative bacteria?

Comments on the Quality of English Language

Moderate editing of English language required

Author Response

It is important to clarify in the evaluation of thyme essential oil, why was a larger zone of inhibition generated using 40 μl of essential oil compared to 80 μl? (Figure 1 and 2).

Added statement to manuscript stating the difference was not statistically relevant and was likely due to experimental variation (Page 5, Results section lines 6-8)

As advice for your future investigations, consider possible diffusion problems in the evaluation of the antimicrobial properties of oils and their components. Would the behavior of the oils and their components be like gram-negative bacteria?

We appreciate the advice and will consider this in our future studies.